# Absorption of Toxicants from the Ocular Surface: Potential Applications in Toxicology

**DOI:** 10.3390/biomedicines13030645

**Published:** 2025-03-06

**Authors:** Ivan Šoša, Manuela Perković, Ivanka Baniček Šoša, Petra Grubešić, Dijana Tomić Linšak, Ines Strenja

**Affiliations:** 1Department of Anatomy, Faculty of Medicine, University of Rijeka, 51000 Rijeka, Croatia; 2Department of Pathology and Cytology, Pula General Hospital, 52100 Pula, Croatia; manuela.perkovic@gmail.com; 3Clinical Hospital Centre Rijeka, University Department of Physical and Rehabilitation Medicine, Krešimirova 42, 51000 Rijeka, Croatia; ivanabanicek@net.hr; 4Department of Ophthalmology, Clinical Hospital Center Rijeka, Krešmirova 42, 51000 Rijeka, Croatia; grubesic.petra@gmail.com; 5Department for Health Ecology, Faculty of Medicine, University of Rijeka, Braće Branchetta 20, 51000 Rijeka, Croatia; dijanatl@uniri.hr; 6Department for Scientific and Teaching Activity, Teaching Institute of Public Health County of Primorje-Gorski Kotar, Krešimirova 52a, 51000 Rijeka, Croatia; 7Department of Neurology University Hospital Centre Rijeka, Faculty of Medicine, University of Rijeka, 51000 Rijeka, Croatia; medines4@yahoo.com

**Keywords:** drug, eye, matrix, modeled tissue, particle pollution, ocular surface fluid, toxicants, toxicology

## Abstract

In relation to the eye, the body can absorb substances from the ocular surface fluid (OSF) in a few ways: directly through the conjunctival sac, through the nasal mucosa as the fluid drains into the nose, or through ingestion. Regardless of the absorption method, fluid from the conjunctival sac should be used as a toxicological matrix, even though only small quantities are needed. Contemporary analytical techniques make it a suitable matrix for toxicological research. Analyzing small quantities of the matrix and nano-quantities of the analyte requires high-cost, sophisticated tools, which is particularly relevant in the high-throughput environment of new drug or cosmetics testing. Environmental toxicology also presents a challenge, as many pollutants can enter the system using the same ocular surface route. A review of the existing literature was conducted to assess potential applications in clinical and forensic toxicology related to the absorption of toxicants from the ocular surface. The selection of the studies used in this review aimed to identify new, more efficient, and cost-effective analytical technology and diagnostic methods.

## 1. Introduction

While the first-pass effect is often associated with the liver as a significant site of drug metabolism, it is a pharmacological phenomenon in which a medication undergoes metabolism at a specific location in the body prior to reaching its endpoint. Thus, the first-pass effect mitigates the effect of the medication itself, decreasing the active concentration.

It has been demonstrated in numerous experiments and publications that fluid from the ocular surface ultimately (at least in minute amounts) enters systemic circulation due to passive diffusion and the anatomy of the lacrimal output [1], via the nasal/pharyngeal mucosa by either ocular venous or lymphatic drainage, which is how xenobiotics from the ocular surface fluid (OSF) reach systemic blood flow [2,3] (Figure 1).

Passive diffusion along the concentration gradient is the primary method of transport for xenobiotics in the OSF (Figure 1) [6]. Regarding the construct of xenobiotics, the standard definition of “chemicals found but not produced in organisms or the environment” is adopted in this paper [7].

Since passive diffusion is a bidirectional process, even the secretion of drugs in tears has been investigated for a limited number of drugs. The transfer of the free non-protein-bound fraction from plasma to tears can only be expected for compounded drugs that exhibit sufficient lipid solubility. For acidic and essential drugs, the correlation of the tear-to-unbound plasma ratio is disturbed by variations in tear pH, a value used to specify the acidity or basicity of aqueous solutions [8]. A measure of an acid’s strength is pKa; the higher the pKa, the weaker the acid, and the lower the pKa, the stronger the acid. Similarly, the weaker the acid, the stronger its conjugate base, and the stronger the acid, the weaker its conjugate base.

This review acknowledges the limitations of traditional toxicology testing platforms, including ethical issues, high-cost animal models with restricted relevance to humans, low throughput, and inconsistent responses. It therefore introduces alternative methods to toxicology, i.e., in vitro tissue models, engineered tissues, and OSF.

### The Principle of 3Rs (Reduction, Refinement, and Replacement)

A small amount of fluid in the space in front of the eye or the area between the eyelid and the eyeball is ideal for applying the 3Rs (reduction, refinement, and replacement). This principle was developed over 60 years ago [9] and now serves as the foundation of analytical toxicology, which is integrated into current regulatory standards (although Russell et al. replaced humans with animals, and in our model, conventional matrices are replaced with OSF). It provides a framework for conducting more humane research involving humans and animals. The 3Rs concept supports a legal and ethical framework for in vivo research [10]. According to Directive 2010/63/EU [11], this principle must be considered when selecting testing methods for the regulatory assessment of human and veterinary medicinal products.

While this principle was initially designed to replace human experiments with those conducted on animals, utilizing OSF and in vitro models can facilitate a significant shift in methodology.

This review aims to demonstrate how toxic molecules are absorbed from the ocular surface, with potential applications of this feature in toxicology, as related to diagnostic and industrial issues [12].

## 2. Literature Review

A search of scientific literature was conducted across three databases: PubMed, Scopus, and the Web of Science Core Collection. The search, performed using a web browser on 2 December 2024, aimed to identify all studies containing the terms “in vitro model” AND “forensic toxicology” in any of the search fields. The assessment included studies from the inception of each database. Automated software identified three duplicates among the initially identified 158 studies. Additionally, 56 studies were found ineligible due to being secondary (desk) research or classified as “gray literature.”

Ultimately, 93 studies were included in the screening process, of which 20 were excluded upon detailed review because their topics and content were not related to humans/full texts were not in English. This left 73 studies that underwent an in-depth screening after we obtained the full texts. A significant number of these full texts were available as open access (*n* = 49; 67.12%), while others were retrieved using the built-in tool in EndNote (*n* = 10; 12.65%) from online repositories (*n* = 6; 8.22%). Despite these efforts, full texts could not be retrieved for eight studies (11%). The entire process of study identification is illustrated in the diagram in Figure 2.

## 3. Matrix

The OSF usually coats the surface of the conjunctiva in the conjunctival pocket, and most of the fluid (approximately 80%) introduced to that pocket overflows. A barely significant amount of 5–10% of the OSF reaches the pharyngeal mucosa, and another 5–10% of the OSF penetrates to deeper ocular structures; both outputs eventually reach systemic circulation [13]. Only a tiny portion of the OSF that reaches the pharyngeal mucosa will be swallowed and subjected to first-pass liver metabolism [14,15]. The conjunctiva forms a sac with the corneal epithelium, creating the core ocular surface. This results in a continuous wettable lining of the conjunctival sac, which opens anteriorly [16]. The broader definition of the ocular surface includes the cornea, conjunctiva, lacrimal glands, and associated lid structures [17,18,19]. This concept has other significant offshoots, including a lacrimal functional unit and an ocular surface system.

The lacrimal functional unit may seem broad, as it includes all of the structures responsible for maintaining the tear film and protecting the transparency and health of the ocular surface. On the other hand, the ocular surface system is considered to be the continuous epithelia of the ocular surface, which is connected through ductal epithelia with the acinar epithelia of the principal and accessory lacrimal glands and other supporting glands, as well as the nasolacrimal system. Regardless of our conceptualization of that space, it contains OSF, a fluid enriched by a mucoaqueous pool (MAP). This fluid is enclosed and secluded from the atmosphere and evaporates into the three-dimensional ocular surface sac sealed by the ever-present lipid sealant. The MAP acts as a lubricant during eyelid and eyeball movements. Meibomian oil lipids make up the central part of the lipid sealant layer, and they do not enter the conjunctival sac [20]. The liquid part of the ocular surface system coats and wets all parts of the conjunctival sac and plays a role in optical functions [21].

The external corneal surface protrudes to the outer environment, forming a functional interface with the inner ocular structures. Anteriorly convex and transparent, it is the foremost part of the eyeball’s outer layer. It protrudes above the level of the sclera, and it is more curved than the sclera. For that reason, it extends to a dome-shaped elevation with an area of 1.1 cm^2^ [22]. Antero-posteriorly arranged and viewed under the microscope, the cornea consists of five layers: the corneal epithelium, the anterior limiting lamina (Bowman’s layer), the substantia propria (stroma), the posterior limiting lamina (Descemet’s membrane), and the endothelium of the anterior chamber. The corneal epithelium, which makes up approximately 10% of the corneal thickness (50 μm), consists of 5–6 outermost layers of cells. This layer serves to protect the ocular surface from mechanical abrasion; forms a permeability barrier for small molecules, water, and ions; and prevents the entry of pathogens [23,24]. Consistent with their barrier function, 2–3 superficial layers of wing-shaped cells display extensive protrusions organized in a complex network of tight junctions [25] up to the junction of the cornea with the sclera, a thin, transparent mucous lining that swerves from the lids to form fornixes (pockets) of varying depths (Figure 3).

These so-called conjunctival pockets were measured from the upper eyelid and reflected onto the globe covering the sclera, with a size of 15.6 mm (95% CI [12.5–18.8 mm]). The lower conjunctival fornix depth was measured as 10.9 mm (95% CI [ 8.0–13.7 mm]) by Jutley et al. in 2016 [26]. Organoids exhibit all of their comparative advantages in this context.

Medication is often administered to this pocket by turning the eyelid inside out and injecting it under the inner lining. Depending on the amount injected, a small part of the medication may leak from the injection site onto the eye’s surface and mix with the OSF [27].

Whether the surface of the functional interface with the inner ocular structures is called a lacrimal functional unit or an ocular surface system, it represents a corneo-conjunctival surface interacting with the environment. Portraying the eye as mainly resistant to external influences, in anatomical and physiological terms, this unique barrier can be considered a precorneal part of the corneal barrier. Its precorneal clearance mechanism refers to the tearing process and reflex blinking [28]. An overflow of excess liquid from this confined space causes fluid to be expelled either onto the skin or through the nasolacrimal duct (a canal of 265.33 ± 90.57 mm^3^ in men) [29,30].

### Methods of Fluid Collection

While blood is the most commonly used body fluid, tears are less complex and more easily accessible for collection. The volume of individual samples is limited to 10–15 µL from each eye at a time. Today, the concept of tear film considers the transparent three-layered fluid covering the eye’s surface, which is primarily under-researched, undervalued, and ignored as a toxicological matrix. However, it shows promise as an alternative to traditional fluids in analytical toxicology [31,32]. As described in the work of Zhou et al. (2012) [33], tear samples can be collected after ensuring that subjects are not wearing contact lenses. The sampling can be performed using a standard Schirmer’s strip, which is a conventional tool for this purpose. These strips must be frozen immediately after collection and kept frozen until analysis.

In addition to Schirmer’s strips, earlier research techniques, such as using sponges for conjunctival swabs, were also employed. Some researchers use capillary tubes made of plastic rather than glass to minimize the risk of injury. However, this method can be time-consuming and requires expertise [34]. Specifically, Schirmer’s strips are diced and fully immersed in an elution buffer comprising 100 mM ammonium bicarbonate and protease inhibitor. Barmada and Shippy describe collecting the OS fluid via capillary action on phenol red thread. In their experiment, a color change length indicated the volume of fluid collected [35].

Currently, the recommended devices for fluid collection from the ocular surface (OS) include capillary tubes and Schirmer’s strips. These methods do not require stimulation and are designed to collect volumes suitable for further laboratory analysis, depending on the specific needs of each case and the convenience of the person interpreting the results. In the cohort studied by Bachhuber et al. [34], both methods were found to be safe and well tolerated. For example, Yao et al. [36] recommend using Schirmer’s strips, even for microsampling in the mass spectrometry analysis of human tears, to identify drugs of abuse (Table 1). The subsequent analysis is conducted according to various protocols that can influence the composition and quality of the obtained samples. This variability introduces significant differences among studies.

## 4. Toxicant

Aside from water, salts, and other components (e.g., antibodies) contained in filtrates of blood plasma, OSF comprises lacrimal gland epithelial cells, any medications instilled into the conjunctival sac, and any toxicants from the environment [50]. In fact, the eyes can absorb chemicals, usually when the chemicals are introduced as eyedrops. Alternatively, toxicants can reach the eye surface when they spill or splash onto unprotected eyes [51]. Another possibility is the heavy burden of industrial and transportation pollution, which lessens the quality of the air, reduces the resilience of tear film, and alters the permeability of the eye’s barrier against the environment [52].

## 5. Interface with the Organism

When awake, the eyelid margins are typically separated, exposing the bulbar surface. This surface is coated by an osmotic fluid enriched with MAP, which is a natural part of the eye.

Environmental toxins cause oxidative damage and inflammation at the cellular level. This can impair the motility of the corneal and conjunctival cells and cause their senescence or even apoptosis. In this way, significant tissue barriers that limit ocular drug absorption are infringed [53].

Several studies link meteorological and environmental conditions and the eye surface [54,55,56]; air pollution can mitigate the thickness of the tear film lipid layer, as demonstrated by volunteers in the study by Wang et al. [57]. Unstable tear film was correlated with the air quality index during three years of that study in a convincing regression model, with the following parameters: multiple R = 0.82; *p*-value = 0.00; and a goodness of fit of R^2^ = 0.68. The same was true for the study of Hao et al. [58], which confirmed the presumptions of Torricelli et al. [35] regarding the adverse effects of air pollution on the eye surface. Admittedly, Hao et al. observed the impairment of the meibomian gland and the upregulation of tear cytokine concentrations.

In our literature review, only a non-Latin letter study by Ma et al. [59] initially met the criteria for inclusion. However, a deeper examination of that study suggested its elimination. Additionally, fine particles found in polluted air, such as micro- and nanoplastics, can cause systemic toxicity by penetrating cell membranes [60,61]. There is copious evidence that particle pollutants cause a number of diseases or influence the existing maladies. Plastic particle pollutants are reported as risk factors for cardiovascular diseases and stroke. They are implicated the inetiopathogenesis of developmental disruptions, autism, and attention deficit hyperactivity disorder (ADHD) [62,63,64]. Plastic, particularly pollutants from the environment, precipitates a series of mental disorders. Additionally, plastic debris from environmental pollution is associated with cognitive decline in Alzheimer’s disease [65].

Thus, pollution is associated with causes of systemic inflammatory response, cytokine production [52,66], and vascular inflammation [67]. While the proper role of plastic waste, including substitutes for conventional non-degradable plastic polymers in endothelial quintessence, still needs clarification, Millen et al. [68] performed a scoping review of the scholarly literature to determine whether ambient air pollution was a risk factor for chronic disease of the inner ocular structures and elevated intraocular pressure. Looking at 27 identified articles in which air pollutants were considered responsible for eye conditions, the systemic effects of airborne pollutants entering the system via the ocular surface do not need further discussion. The lack of literature on this topic seems appropriate, but generally, it suggests it is an area requiring further research.

Ongoing tests of environmental pollutants and their harmful effects use organoids and organoids-on-a-chip, taking advantage of their benefits and addressing the limitations of standard approaches. However, this method is still in the initial stage and shows excellent potential for environmental toxicology research [69].

## 6. Medications

Ocular toxicology refers to the harmful effects of drugs administered topically, intraocularly, or systemically. It also encompasses the assessment of adverse effects caused by ophthalmic devices, including contact lenses, intraocular lenses, and glaucoma implants. The primary research eligible for our literature search is listed in Table 2.

Due to various physiological and anatomical constraints, only a small percentage of topically administered medication doses can be absorbed when applied via the eye. It is expected that 99% of the drug is lost from the precorneal area [70].

**Table 2 biomedicines-13-00645-t002:** Studies resulting from our literature search containing the term “medications” in any of the search fields.

Study	Type of Study	Intervention	Outcome	
Abd-Elhakim, Y.M. [71]	Animal	Tartrazine and chlorophyll in rats	Serum levels of immunoglobulins, levels of expression of genes containing interleukins, enzyme-linked immunoassay	
Alqaissy, W.Q.M. [72]	Animal	Treatment of infections of the urinary tract in rats induced by pathogenic *E. coli*		
Gameli, P.S. [73]	In silico metabolite prediction	Metabolism of thieno-triazolo diazepine in human hepatocytes	Web-based in silico prediction	High-resolution mass spectrometry
Heo, D. [74]		Metabolism of vardenafil analogs	Toxicity, safety, efficacy, side effects, drug interaction, and metabolism study	Mass spectrometry and liquid chromatography

It is important to note that the main barrier to the absorption of topically applied drugs is the relatively impermeable cornea, approximately 500 µm thick, lined up with the lipophilic epithelium with tight intercellular junctions. Conversely, the corneal stroma is a highly hydrophilic layer that acts like a liquid and is 1.5 times more viscous than water [75,76]. Drugs pass through this layer passively, using one of the following routes: either between the cells (paracellularly) or through the cells (transcellularly) [77]. Complying with the 3R principle is essential whenever using experimental animals and different physiological microenvironments. Moreover, in the case of accidentally ingested OSF contaminated with environmental pollutants, animal models and two-dimensional (2D) cell lines are unable to accurately simulate the adverse effects of harmful environmental pollutants on humans. For this reason, this review aims to establish the role of tissue models in toxicology (clinical and forensic) relative to the OSF.

While this constantly changes with new approaches to drug delivery, for the time being, the maximum amount of the drug is absorbed into systemic circulation via the conjunctival membrane and the nasolacrimal drainage system, which is marked by constrained bioavailability [78]. In ocular pharmaceuticals, this is a significant concern associated with drug delivery. Alternative approaches such as nano-/microparticles, nanosuspensions, nano-/microemulsions, liposomes, nano-micelles, and dendrimers may overcome the occurrence of precorneal loss [34].

Some drugs, such as phenobarbital, carbamazepine, methotrexate, 5-fluorouracil, ampicillin, and acetylsalicylate, are known to be excreted into the tears [2]. These drugs can significantly change the quality of the tear film. For example, rifamycin, when excreted into the tears, can turn them orange [79]. Other drugs typically used in ophthalmology, such as eye drops containing the topical anticholinergic antagonist tropicamide, are increasingly used non-medicinally as illicit drugs.

### Route for Illicit Drugs

The legal status of tropicamide varies worldwide, and anecdotal reports have linked its misuse to the former Soviet Union. However, there are policies in place to regulate the use of eye drops containing this substance, especially as recent studies have shown that its misuse extends well beyond Eastern Europe [80,81]. Widespread use and an uncoordinated legal status keep tropicamide in the public spotlight because of falsified prescriptions [82].

Ophthalmic drops, which have various clinical applications, have seen an increase in their nonmedical use. Their rapid onset of effects is somewhat counterbalanced by their relatively short duration. This rise in usage coincides with an increase in opioid addiction and drug-related mortality. Additionally, these drops are referred to as “seven-monthers” because they can take up to seven months to be fatal to a young [81], healthy person. When abused via excessive doses or through routes other than the conjunctiva [83,84,85], tropicamide can cause feelings of euphoria, visual and auditory hallucinations, convulsions, and ataxia. The existence of this drug of abuse, which is administered via the eyes and follows the same route as conventional eye drops, sheds new light on the ocular surface as a route of illegal drug use rather than just for medical treatment. This drug is generally known as a cheap alternative to heroin, and it enhances the favored effects of heroin. Isolating this drug from the OSF is not conclusive evidence of its ocular use and abuse; moreover, passive diffusion is a bidirectional process, although there are numerous reports of intravenous abuse [83,86]. Another ocular medication, tetrahydrozoline eye drops, was discovered in toxicological samples, indicating possible tampering with toxicological urine screening among cannabis users. The presence of the same drug in biological samples may indicate the occurrence of chemical submission [87,88], but in recent years, its abuse has been increasingly linked to lethal outcomes [89,90]. Chemical submission should be suspected even when clonidine is identified in biological matrices [91]. Based on these reports, according to the frequency of dispensing, drugs that are not used in criminal activities are still the most commonly prescribed. Lubricants, antibiotics, steroids, and non-steroidal anti-inflammatory drugs (NSAIDs) account for 62.14% [92].

Any quantity of the OSF swallowed is ultimately subjected to first-pass metabolism due to gastrointestinal mucosa absorption. Blood drained off the mucosa contains a minimal quantity of absorbed xenobiotics from the OSF that pass through the liver. The central role of the liver in this process is absorption, which occurs through the gut wall [93]. This effect is mostly associated with orally administered and intestinally absorbed medications, but it can still occur with drugs delivered in other ways. The absorption of a substance is influenced by factors such as plasma protein concentration, gastrointestinal motility, and enzymatic activity involving body systems such as the lungs, vasculature, gastrointestinal tract, and other metabolically active tissues. Due to the solid first-pass effect, a significantly higher oral dose is often required compared to the dose administered directly into systemic circulation. Marked individual variations in oral doses due to differences in the extent of first-pass metabolism are noted as well [94]. These factors should be considered when determining the appropriate dose or providing expertise regarding the prominence of the first-pass effect regarding OSF.

The liver deactivates xenobiotics and decreases their bioavailability (the share of the amount ingested that finally reaches systemic circulation). However, when medications made of protein that are a part of the OSF are ingested and presented to the gastrointestinal mucosa, enzymes deactivate them as they pass through the stomach and duodenum.

## 7. In Vitro Models

Assays augmented with in vitro tissue models and high-throughput procedures produce robust data. This reduces the frequency of false positives and negatives, allowing the data to be used in computational modeling [95,96]. First, it is important to differentiate between organoids and organs-on-a-chip. Organoids are three-dimensional clusters of cells created by stem cells organizing themselves based on the principles of developmental biology [97]. The most simplified explanation for organoids is that they are 3D in vitro models made up of specific mature cells differentiated from adult stem cells (ASCs), induced pluripotent stem cells (iPSCs), or embryonic stem cells (ESCs). These models mimic real organs and are capable of reproducing certain specific functions in a laboratory setting [98,99]. The concept of organs-on-a-chip is more complex. It refers to a cell culture approach, approximately the size of a USB flash drive, in which different cells are cultured on different levels of the surface [99].

Pathologies or biological phenomena, such as the unintentional swallowing of the OSF, require interactions between several organs (e.g., the eye) and organ systems (e.g., the digestive system). Therefore, they require the use of animals as experimental tools [100]. Organs-on-a-chip, on the other hand, are microfluidic cell culture devices that are typically manufactured as microchips [101].

Only one study regarding forensic toxicology from our literature review mentioned tissue models, specifically genetically modified next-generation liver organoids [102]. These models offer a new approach to personalized medicine in therapeutic efficiency and toxicity studies. Therefore, we concluded that unlike predictive toxicology, in silico techniques, and drug development, clinical and forensic toxicology may not directly benefit from these advancements (Figure 4) [103,104,105,106,107]. In our review, 9 out of 73 studies (1.23%) included the term “in silico”, whereas “in vitro” was used in 37 studies (50.68%).

The throughput of organoid-based assays outweighs that of animal models.

Conducting preclinical studies on animals to predict the toxicity of new drugs or cosmetics and their outcomes in humans is challenging and hindered by inconclusive results [108]. Moreover, the European Union prohibits such animal-based experiments [109]. An evaluation of the adverse effects in humans of 150 drugs using an animal model showed an accuracy of only 71% [110]. Clinical trials revealed that nearly half of the drugs causing liver injury are not identified as harmful by animal models [111]. Even though the drug development process typically takes about a decade, only around 0.02% of candidate drugs ultimately succeed [112,113], and many approved drugs are later withdrawn from the market [114]. This leads to significant financial losses for the pharmaceutical industry and a massive impact on healthcare [107,115]. In addition, technologies like therapeutic CRISPR RNAs or monoclonal antibodies do not work in animal models at all [116].

Generally, using more accurate human liver models reduces animal usage and increases testing efficiency [117], even for medications administered to the eye. This ultimately leads to reduced costs for predictive and environmental toxicology. In the early phases of drug development, this is achieved via cell-based screening of a large number of potentially active molecules against a specific target. The absorption, distribution, metabolism, excretion, and toxicity of a drug in the preclinical stage can be clarified using animal models or modeled tissue.

The metabolism and first-pass effect of drugs can be easily studied using liver organoids or liver-on-a-chip methods. Similarly, developing a full 3D in vitro model of the cornea would significantly support the hypothesis presented in this article. However, there are still several challenges in the field of cell culturing that need to be addressed before this breakthrough can be achieved [118]. The latest in vitro corneal construct developed by Islam et al. [119] offers significant promise, as it features a fully functional model of the human cornea. This model incorporates three types of human corneal cells and neural cells derived from hybrid neuroblastoma cells. The viability and functionality of the construct were evaluated based on the typical organization of cellular layers and the expression of protein profiles characteristic of each specific cell type, confirmed via Western blotting. The presence of typical cellular layers and the maintenance of the individual cell phenotypes make this model valuable for evaluating specific corneal disorders. However, it has not yet been assessed within a toxicological framework. The authors acknowledge its potential for drug targeting and for reducing the reliance on animal models in early-stage corneal research.

The development of databases, computational models, and simulations actively contributes to advances in medicine. Software and virtual chemical spaces can assist in making decisions and predictions and generating hypotheses, potentially minimizing the need for in vivo testing of animals [120]. Subsequently, the structures of candidate drugs are advanced to improve target specificity and selectivity, as well as their pharmacodynamics, pharmacokinetics, and toxicological properties. In the field of pharmacology, in silico methods are extensively used for toxicology tests and drug checking [121], reducing time and cost in regards to drug discovery. The rapid development of in silico technologies indicates a promising future [122,123]. Current in silico methods enhanced by AI-based techniques can help select potential patients for preclinical trials and manage possible toxic or unnecessary side effects [124]. For the time being, in silico models can successfully predict human gut and liver clearance, but additional validation with a wider range of drugs and physiological fluids will be needed. In the context of forensic toxicology, for instance, in silico techniques are applied to quantum chemistry and multivariate analysis of the infrared spectra of new psychoactive substances (NPSs) [125]. The increasing number of NPSs is associated with many issues in the context of law enforcement and public drug policies, including transitioning such experiments to modeled tissues. This creates a feasible landscape for expert toxicological work and drug characterization [126]. The development of analytical procedures is struggling to keep up with the rapid pace at which NPSs are being introduced to the market. This limitation hinders our understanding of the short- and long-term effects of these substances and the risk they pose to consumers. To broaden our knowledge, we can utilize modeled tissues and in silico techniques [127]. Research that is not in silico (i.e., in vitro and in vivo research) requires expensive and time-consuming procedures. Therefore, searching for faster and less costly study alternatives seems reasonable.

## 8. Conclusions

Ocular surface fluid can enter systemic circulation, allowing foreign substances (xenobiotics) to reach the bloodstream through passive diffusion. The first-pass effect occurs when a medication is metabolized in metabolically active tissues before it reaches its target, which decreases its effectiveness. Throughout the day, the eye is exposed to environmental toxins that can damage cells and disrupt the tear film. Additionally, tiny particles of air pollution can thin the lipid layer of the tear film, leading to instability.

In this paper, we discuss the use of OSF for toxicological purposes, both clinical and industrial. We highlight the limitations of animal testing and the potential benefits of utilizing human tissue models and in silico techniques. Engineered tissues and in vitro models are ideal for evaluating the toxic effects of various drugs in a high-throughput setting. Apart from the ecological aspects of pollution that come with industrialization and transportation, high levels of air pollution impair human health, including the homeostasis of the OSF. In that vein, the environmental impact on tear film quality needs to be scrutinized. High-throughput studies conduted for organoids should include experiments applicable in the fields of forensic and clinical toxicology, taking into account the lack of a central policy in the European Union to address drug abuse. For that reason, some controlled substances can easily enter the legal supply chain, and this could be reduced by toxicological assessment of the OSF. A more in-depth study is needed to comprehend the full extent of problems arising from the fact that practically anyone can purchase almost any medication from a pharmacy without a prescription, which has led to the abuse of both over-the-counter and prescription drugs, including ophthalmic preparations. In conclusion, we highlight the need for improved analytical procedures to keep pace with the rapid emergence of new psychoactive substances.

## Figures and Tables

**Figure 1 biomedicines-13-00645-f001:**
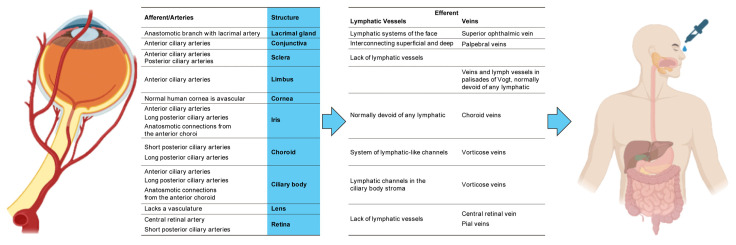
This review focuses on the ocular blood supply and lymphatic drainage. Key structures relevant to this discussion are outlined in the accompanying figure. For instance, the eyelids and pre-extrinsic eye muscles are not included in this outline, and the term “conjunctiva” refers to both the palpebral and bulbar types. The terminology used in this figure aligns with the FIPAT’s International Anatomical Terminology (Terminologia Anatomica—Second Edition) [4,5]. Additionally, ocular surface fluid migrates across a concentration gradient, which can influence the presence of drugs in the ocular surface fluid (OSF), even if those drugs are not administered directly to the eye. (Figure made using BioRender, University of Rijeka, Croatia, https://www.biorender.com, accessed on 17 February 2025, and Microsoft^®^ PowerPoint^®^, Microsoft 365, 64-bit version 16.0.17531.20140, University of Rijeka, Croatia).

**Figure 2 biomedicines-13-00645-f002:**
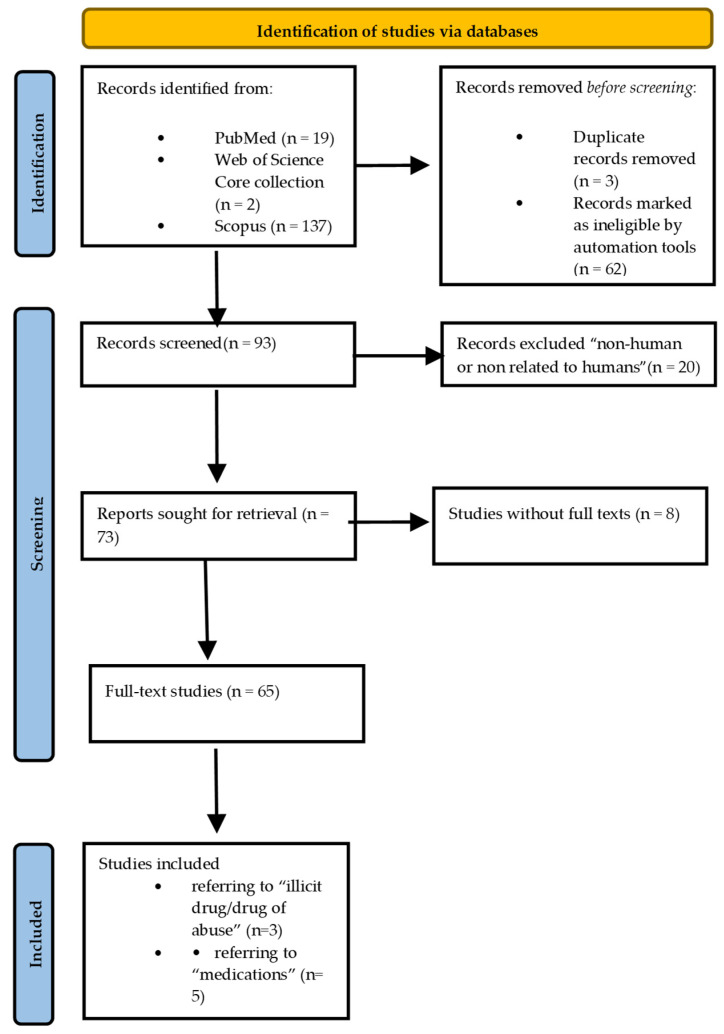
PRISMA 2020 diagram showcasing our search in three levels.

**Figure 3 biomedicines-13-00645-f003:**
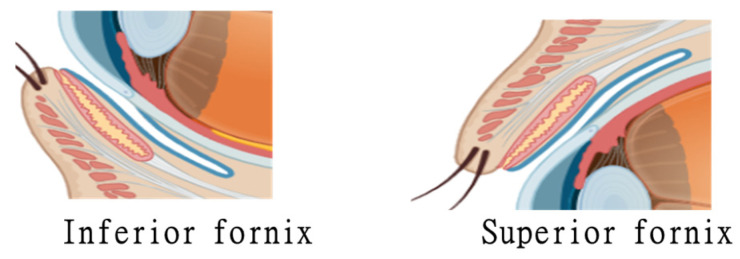
Fornixes (pockets) of the conjunctival sac. (Figure made using BioRender, University of Rijeka, Croatia, https://www.biorender.com, accessed on 17 February 2025).

**Figure 4 biomedicines-13-00645-f004:**
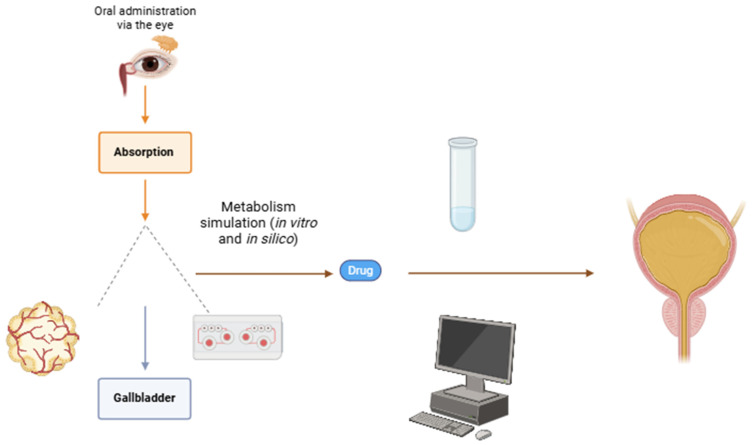
Schematic of a hypothetical pharmacokinetic model for drug metabolism using in vitro/in silico methods, including liver organoids/liver-on-a-chip. The development of liver organoids, organs-on-a-chip, and the least complex spheroids provides a platform for personalized medicine studies as well. Simulations of the gallbladder and urinary bladder are also shown. (Figure made using the BioRender account of the University of Rijeka, Croatia, https://www.biorender.com, accessed on 17 February 2025).

**Table 1 biomedicines-13-00645-t001:** Sampling and analysis methods used for identifying and/or quantifying various analytes in the ocular surface fluid.

Analysis Method	Contaminant	Collection Method	Reference
Agar diffusion assay	Air pollution	Filter paper	Berra et al., 2015 [37]; Galperín et al., 2018 [38]
ELISA ^1^	Tobacco smoke	Capillary tube	Rummenie et al., 2008 [39]
ELISA	Tobacco smoke	Capillary tube	Rummenie et al., 2008 [39]
Ethanol assay kit	Alcohol	Capillary tube	Kim et al., 2012 [40]
GC-MS ^2^	Air pollution	Schirmer strip	Gutierrez et al., 2019 [41]
ICP-MS ^3^	Trace elements	Capillary tube	Chen et al., 2022 [42]
Immunoassay	Mold	Capillary tube	Peltonen et al., 2008 [43]
Immunoassay	Air pollution	Capillary tube	Matsuda et al., 2015 [44]; Jing et al., 2022 [45]
LC-MS ^4^	Ozone	Capillary tube	Paananen et al., 2015 [46]
PIXE ^5^	Air pollution	Schirmer’s strip	Girshevitz et al., 2022 [47]
PSMs ^6^	Smoke	Schirmer’s strip	Yao et al., 2020 [36]
PSMs	Aerosols	Schirmer’s strip	Yao et al., 2020 [36]
PSMs	Drugs of abuse	Schirmer’s strip	Yao et al., 2020 [36]
PSMs	Volatile organic compounds	Schirmer’s strip	Yao et al., 2020 [36]
SEM/EDS ^7^	Particulate matter	Schirmer’s strip	Avula et al., 2017 [48]
SEM/EDS	Indoor environment	Schirmer’s strip	Kaplan et al., 2019 [49]

^1.^ ELISA = enzyme-linked immunosorbent assay; ^2^. GC–MS = gas chromatography–mass spectrometry; ^3^. ICP–MS = inductively coupled plasma–mass spectrometry; ^4^. LC–MS = liquid chromatography–mass spectrometry; ^5^. PIXE = particle-induced X-ray emission; ^6^. PSMs = problem structuring methods; ^7.^ SEM/EDS = scanning electron microscopy (SEM) and energy dispersive X-ray spectroscopy (EDS).

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
