# Peer review of "Absorption of Toxicants from the Ocular Surface: Potential Applications in Toxicology"

_biomedicines, 2025, doi:10.3390/biomedicines13030645_

Round 1

Reviewer 1 Report

Comments and Suggestions for Authors

Dear authors,

Thank your effort, the work covers an overlooked topic that can be addressed.

I would you revise your manuscript to meet the standard of scientific writing, also I would recommend revising the listed refrences and include labarotry experiment details and condition, compare the result in deeper scientific structure, you need to re write the structure of  manuscript, 

Here are few examples :

Lines 41 -42 space

Line 45 space

Line 46 pr is misplaced

Line 87 use formal scientific language please

Figure 1 needs illustrations of the stream described in test

Lines 88-90 rephrase please

Lines 99-100 needs elaboration and rephrasing

Lin2 111 typo, use superscript please

If I wrote this manuscript, I would place matrix before the objective of the study

Line 143 grammar and language

Is it proper to use the term grey literature in publication?

Line 185 superscript when appropriate, revise text please

Line 214 definition of ocular toxicology does not seem scientific. This is not the description of toxicology

Table 2, levels of expression of genes contzoling interleukins, what is contzoling?

Line 247: Phrase with some drug can be written in scientific soundness

Line 260 scientific writing

Comments on the Quality of English Language

English as language and scientific writing needs monitoring

Author Response

Dear reviewer, thank you for your comments.

  1. We have improved our language skills and have now provided an English language certificate. The presentation style has been enhanced, and typos have been eliminated.
  2. If I wrote this manuscript, I would place matrix before the objective of the study. The manuscript has been thoroughly reorganized; the "matrix" section has been moved forward, and the objective has been verbalized earlier.
  3. Laboratory work (sampling and analysis) has been added, but please note that one of the authors authored an article on a similar topic: https://doi.org/10.3390/toxics12070513.

Reviewer 2 Report

Comments and Suggestions for Authors

Please, find attached the file with all the details.

Comments on the Quality of English Language

The level of English is sometimes poor and makes reading difficult. 

Author Response

Dear Reviewer,

Thank you for your thoughtful comments.

We have revised our title and abstract accordingly.

  1. In response to your comment on Page 3, line 76, regarding the limitations of classical toxicology tests, we have specified these limitations as requested.
  2. We also appreciated your feedback on Figure 4. We have removed Figures 3 and 4, as they were determined to be of low quality and did not contribute meaningfully to the text.
  3. Regarding your observation about the abstract: we agree that the first part was unclear. We have reorganized the entire article to improve clarity, removed references to the 3Rs from the abstract, and introduced a specific section in the paper discussing this principle. Additionally, we have changed the title to better reflect this direction.

Thank you again for your valuable input.

Reviewer 3 Report

Comments and Suggestions for Authors

1)      It is recommended that the ocular blood supply and lymphatic drainage be presented in a schematic diagram, as this can provide the reader with better clarity.

2)      Lines 126 and 127: What does “CI” stand for?

3)      Explain more about “They are implicated in developmental disruptions, autism, and attention deficit hyperactivity disorder (ADHD)”. Does this mean during embryonic development in the pregnant term?

4)      It can be useful to summarize literature in one or two tables.

5)      (Figure 5) is missing. Please check and correct.

6)      Line 327: “The throughput of organoid-based assays outweighs that of animal models.” How did the authors compare? What were the comparison criteria? Please mention in the literature.

7)      Describe “CRISPR” in a footnote.

8)      Please explain more about the study by Islam et al.

9)      Some of the issues discussed in the text are unrelated to the study's objectives. For example, “A more in-depth study is needed to comprehend the full extent of problems arising from the fact that practically anyone can purchase almost any medication from a pharmacy without a prescription, which has led to the abuse of both over-the-counter (OTC) and prescription drugs, including ophthalmic preparations.” It is recommended that irrelevant content be removed from the text.

10)  It is recommended that the title be changed to cover the study objectives.

11)  The text requires a complete review where the objectives are clear, which are clarified in the study's introduction. The conclusion should refer to the same objectives and summarize them. This coherence is not seen in the text and the different parts.

Author Response

Dear Reviewer,

  1. We appreciate your comment regarding the recommendation to present the ocular blood supply and lymphatic drainage in a schematic diagram, as this could enhance clarity for the reader. In response, we have combined Figure 1 and Table 1 into a single figure.
  2. Regarding the 95% Confidence Interval (CI), we acknowledge that it is a common statistical value. However, we did not provide the full definition, as it is not an abbreviation specific to this manuscript.
  3. Additionally, we have summarized the literature on sampling and laboratory work in a table.

Thank you for your feedback.

Round 2

Reviewer 1 Report

Comments and Suggestions for Authors

Thank you

Nor further comments

Author Response

Many thanks for all your commemts. 

Reviewer 3 Report

Comments and Suggestions for Authors

 -        The title should be revised to reflect the topic and objectives better. 

-        Fig1:  The writing lacks clarity and needs to be reviewed.

-        There is a lack of coherence between the title, abstract, and conclusion, leading to the inclusion of unrelated content. The authors should clearly state the objectives in the introduction and ensure that the title, abstract, and conclusion are aligned with these objectives.

Author Response

Thank you for your comments. Figure 1 has been updated by the professional editing service (MPI Author Services). The title and objectives have been revised to better align with the article’s content.